# Bead-Containing Superhydrophobic Nanofiber Membrane for Membrane Distillation

**DOI:** 10.3390/membranes14060120

**Published:** 2024-05-23

**Authors:** Md Eman Talukder, Md. Romon Talukder, Md. Nahid Pervez, Hongchen Song, Vincenzo Naddeo

**Affiliations:** 1Department of Physical Chemistry and Physical Chemistry of Polymers, Faculty of Chemistry, Nicolaus Copernicus University, 87-100 Toruń, Poland; 2Guangdong Key Lab of Membrane Material and Membrane Separation, Guangzhou Institute of Advanced Technology, Guangzhou 511458, China; 3Shenzhen Institute of Advanced Technology, Chinese Academy of Sciences, Shenzhen 518055, China; 4Department of Chemistry, Government Saadat College, Tangail, Dhaka 1903, Bangladesh; mrtalukder2002@gmail.com; 5Sanitary Environmental Engineering Division (SEED), Department of Civil Engineering, University of Salerno, 84084 Fisciano, Italy; perveznahidmd@gmail.com (M.N.P.); vnaddeo@unisa.it (V.N.)

**Keywords:** polymer, electrospinning, bead nanofiber membrane, superhydrophobicity, membrane distillation, salt rejection

## Abstract

This study introduces an innovative approach to enhancing membrane distillation (MD) performance by developing bead-containing superhydrophobic sulfonated polyethersulfone (SPES) nanofibers with S-MWCNTs. By leveraging SPES’s inherent hydrophobicity and thermal stability, combined with a nanostructured fibrous configuration, we engineered beads designed to optimize the MD process for water purification applications. Here, oxidized hydrophobic S-MWCNTs were dispersed in a SPES solution at concentrations of 0.5% and 1.0% by weight. These bead membranes are fabricated using a novel electrospinning technique, followed by a post-treatment with the hydrophobic polyfluorinated grafting agent to augment nanofiber membrane surface properties, thereby achieving superhydrophobicity with a water contact angle (WCA) of 145 ± 2° and a higher surface roughness of 512 nm. The enhanced membrane demonstrated a water flux of 87.3 Lm^−2^ h^−1^ and achieved nearly 99% salt rejection efficiency at room temperature, using a 3 wt% sodium chloride (NaCl) solution as the feed. The results highlight the potential of superhydrophobic SPES nanofiber beads in revolutionizing MD technology, offering a scalable, efficient, and robust membrane for salt rejection.

## 1. Introduction

Membrane distillation (MD), a process facilitated by thermal energy and involving membranes, has emerged as a practical and cost-effective solution compared to traditional desalination methods like multi-stage flash vaporization. Its advantages include straightforward operation and the capability to employ low-quality heat sources, including waste heat and solar power. In contrast to pressure-driven desalination methods like reverse osmosis, the MD process potentially offers reduced energy requirements and a lesser propensity for membrane fouling [1,2,3,4]. Nevertheless, the primary challenge hindering the widespread adoption of the MD process in desalination is the absence of efficient membranes that simultaneously exhibit high vapor permeability, effective salt rejection, robust thermal stability, and excellent anti-fouling properties [5,6,7,8].

Recently, the bead nanofiber membrane, characterized by its asymmetrical surface wettability on each side, has garnered significant interest for its promising application prospects in the direct contact membrane distillation (DCMD) process [9,10,11]. The bead nanofiber membrane’s thick hydrophilic layer enhances thermal resistance while concurrently reducing the distance for vapor transfer in the DCMD process [12,13,14,15,16]. Conversely, its thin hydrophobic layer serves as a separation barrier between the feed and cooling solutions, permitting only the passage of water vapor through the membrane. Hence, bead nanofiber membranes can attain greater water flux without sacrificing salt rejection efficiency, unlike conventional hydrophobic membranes [17,18,19,20].

Electrospinning stands out as the preferred method for producing bead nanofiber membranes among existing fabrication techniques, thanks to its suitability for large-scale production, precise control over membrane microstructure and characteristics (such as high surface area, high roughness value, thickness, high porosity, and high hydrophobicity), and the straightforward integration of additional functional materials [2,9,21,22,23]. Recently, considerable research efforts have been focused on developing new membranes with high salt rejection efficiency for extended usage. Despite significant advancements in crafting superhydrophobic electrospun membranes for MD applications, challenges remain regarding their durability and robustness, mechanical strength, and ease of manufacturing [24,25,26,27,28]. Recent studies have highlighted that nanofiber membranes or composites incorporating nanoparticles, carbon materials, and bead-formation nanofibers can significantly enhance nanofiber membrane characteristics and performance [29,30,31,32,33]. Nevertheless, bead formation of nanofiber membranes is being rigorously investigated, either as independent materials or as fillers in polymer composites, for various applications [2,9,22,34,35]. Nanofiber beads are regarded as excellent nanofillers due to their high aspect ratio, substantial thermal and mechanical stability, and lightweight characteristics [17,36].

The formation of beads in the fabrication of sulfonated polyethersulfone (SPES) nanofibers is a phenomenon observed during the electrospinning process, which can significantly influence the morphology and performance of the resultant nanofibrous structures [2,12]. This occurrence is primarily attributed to the complex interplay of solution properties, electrospinning parameters, and environmental conditions [11,22,29,37]. Bead formation often results from the viscoelastic properties of the SPES solution, wherein an insufficient polymer concentration or high surface tension prevents the formation of a stable jet during electrospinning, leading to the formation of spherical structures or beads along the nanofiber. Additionally, electrospinning parameters such as applied voltage, flow rate, and the distance between the needle and collector can further impact bead formation [38,39,40]. Optimizing these parameters is crucial for achieving bead-free SPES nanofibers with uniform diameters and improved surface area, which are essential for applications ranging from filtration to tissue engineering [41,42,43]. Understanding the underlying mechanisms of bead formation and adjusting the electrospinning process accordingly is vital for developing high-quality SPES nanofibrous materials for membrane filtration [2,11,44]. Integrating nanobeads into the nanofiber membrane matrix could alter its properties, potentially enhancing the performance characteristics of the membrane when utilized in MD processes [45,46].

SPES nanofiber membranes have been extensively researched for heavy metal removal due to their exceptional thermal and mechanical properties as well as their chemical resistance [2,47,48]. However, an SPES nanofiber membrane may not be ideal for certain filtration experiments due to a few key limitations. The increased hydrophilicity, while beneficial for water permeability, can lead to issues such as excessive swelling, reduced mechanical stability, and enhanced fouling susceptibility, compromising membrane integrity and efficiency [49]. Therefore, several methods have been employed to improve nanofiber membrane strength, such as surface modifications, particle incorporation, and graft copolymerization. On the other hand, multiwalled carbon nanotubes (MWCNTs) are utilized in membrane filtration due to their exceptional mechanical strength, high chemical stability, and remarkable surface area, which enhance membrane durability, efficiency, and selectivity [50,51,52,53]. Their nanoscale structure improves water permeability while effectively blocking larger contaminants. Additionally, the surface of MWCNTs can be chemically modified to target specific pollutants, and their potential antimicrobial properties help reduce biofouling, making them ideal for advanced filtration applications in water treatment, gas separation, and more. The inclusion of MWCNTs in polymeric solutions has been reported to significantly enhance filtration efficiency [54,55].

Here, oxidized hydrophobic S-MWCNTs were incorporated into the SPES solution through dispersion in concentrations of 0.5% and 1.0% by weight. MWCNTs at 0.5% and 1% concentrations were considered optimum for membrane filtration due to their unique properties [50,56,57,58]. At these concentrations, MWCNTs effectively improve the hydrophilicity of membranes, leading to increased water flux and decreased fouling without compromising the structural integrity of the membrane. Higher concentrations might lead to agglomeration and pore clogging, reducing efficiency, while lower concentrations may not provide significant enhancements [51,53,59]. Thus, 0.5% and 1% are optimal for balancing performance with cost and material handling.

In the presented study, a novel category of SPES@S-MWCNTs bead nanofiber membranes made from SPES polymer, designed for MD, was developed using a co-electrospinning apparatus. This apparatus features a needle capable of processing SPES solutions with varying concentrations of S-MWCNTs, enabling the production of SPES bead nanofiber membranes that consist of two distinct types of nanofibers (pure SPES membrane and bead-formation SPES@S-MWCNTs, each exhibiting unique morphological characteristics. Adjusting the ratios of polymer solution solvents and the operational parameters in the fabrication process altered the traditional nanofiber morphology of SPES nanofiber membranes to a beaded structure with micron-sized beads. The density and shape of the beads in the membranes were examined using scanning electron microscopy (SEM), Fourier transform infrared spectroscopy (FTIR), X-ray diffraction (XRD), and tensile strength, alongside evaluations of WCA. The membranes’ efficacy was evaluated in DCMD mode; water flux and salt rejection were measured, and the membranes’ performance was benchmarked against another nanofiber membrane.

## 2. Materials and Methods

Dimethyl sulfoxide (DMSO) (CAS 67-68-5) was purchased from TNJ Chemical Industry Co., Ltd., Hefei, China. Sulfonated polyethersulfone, SPES (5% sulfonation), was purchased from Changzhou Kete Chemical Co., Ltd., Changzhou, China. Multiwalled carbon nanotubes (MWCNTs), CAS No. 308068-56-6, >90% carbon basis, D × L 110–170 nm × 5–9 μm, were purchased from Sigma-Aldrich, Darmstadt, Germany. Ethanol, CAS: 64-17-5, (anhydrous, ≤0.005% water); sodium chloride, CAS: 7647-14-5; phosphoric acid (H_3_PO_4_), CAS No. 7664-38-2; sulfuric acid (H_2_SO_4_) (95–98%), CAS No. 7664-93-9; and 1H,1H,2H,2H-perfluorooctyltriethoxysilane (CAS No. 51851-37-7) were purchased from Sigma-Aldrich, Shanghai, China. Herein, all polymers and additives were used without further purification steps.

### 2.1. Oxidation Process of MWCNTs

Oxidizing multiwalled carbon nanotubes (MWCNTs) using a concentrated blend of phosphoric acid (H_3_PO_4_) and sulfuric acid (often in a 3:1 ratio) led to the formation of hydroxyl (–OH) groups. These groups enhance the polarity of the MWCNT surface. Additionally, the oxidation process effectively removed metallic contaminants. Specifically, the MWCNTs underwent a 20 h treatment with concentrated phosphoric acid (H_3_PO_4_) and sulfuric acid at 60 °C, which was then followed by filtration through a PET filter and rinsing with distilled water until achieving a pH range of approximately neutral pH.

Following the oxidization of polar MWCNTs and thorough characterization processes, the oxidized MWCNTs were subjected to modification using the chosen 1H,1H,2H,2H-perfluorooctyltriethoxysilane. The salinization process (illustrated in Figure 1) occurred in an argon-filled glove box to maintain a moisture-free environment. The membranes ready for modification were soaked in a 0.1 M solution of the grafting agent for 3 h, forming functionalized nanofiber membranes.

### 2.2. Electrospun Membrane Fabrication

The production of SPES ENMs was carried out utilizing the electrospinning method. A pristine SPES solution was prepared by dissolving SPES at a concentration of 12% (*w*/*v*) in 86 mL of DMSO and stirring the mixture for 4 h at ambient temperature. Hydrophobic S-MWCNTs were incorporated into the SPES solution through dispersion in concentrations of 0.5% and 1.0% by weight. MWCNTs at 0.5% and 1% concentrations were considered optimum for membrane filtration due to their unique properties. At these concentrations, MWCNTs effectively improve the hydrophilicity of membranes, leading to increased water flux and decreased fouling without compromising the structural integrity of the membrane. Higher concentrations might lead to agglomeration and pore clogging, reducing efficiency, while lower concentrations may not provide significant enhancements. Thus, 0.5% and 1% are optimal for balancing performance with cost and material handling.

This mixture was stirred magnetically for about 3 h to achieve a uniform solution. Using an electrospinning apparatus (Model: M06, Foshan Lepton Precision Measurement And Control Technology Co., Ltd., Foshan, China), ENMs were fabricated on a PET nonwoven fabric positioned atop a rotating drum collector, with the process extending for a minimum of 2 h. The solutions, contained in 20 mL syringes, were extruded at a rate of 0.8 mL/h, with an operational voltage of 16 kV at a temperature of 28 °C. The gap between the needle tip and the rotating drum collector was maintained at 18 cm. The thickness of the resultant ENMs ranged from 0.4 mm to 0.5 mm. These bead ENMs were then detached from the PET nonwoven base, and residual solvent was eliminated by drying them in a vacuum oven at 60 °C for 5 h.

### 2.3. Membrane Characterization

The viscosity of the membrane solution at ambient temperature was assessed using a rotational viscometer (NDJ-8S Digital Viscosity Meter, Movel Scientific Instrument Co., Ltd., Ningbo, China). A CAM 200 KSV (Finland) contact angle measurement device was utilized to evaluate the wettability of the fabricated membranes. The data presented were obtained from five distinct readings for each specimen. The morphology of the membranes was examined using scanning electron microscopy (SEM) (Phenom XL, Phenom-World, ThermoScientific, Tokyo, Japan) with an accelerating voltage of 5 kV. X-ray diffraction (XRD) analysis was conducted to further elucidate the structure of the membrane (Empyrean X-ray Diffractometer, Malvern PANalytical), with a 2θ range from 100 to 800° at a 5°/min sweep rate. Functional groups within the porous membrane were analyzed using a Fourier transform infrared spectrometer (FTIR) (Model: Interspec 200-X, Interspectrum, Tartumaa, Estonia), covering a wavelength spectrum from 500 to 4000 cm^−1^ [2].

### 2.4. Direct Contact Membrane Distillation Set-Up

To assess the performance of MD, a lab-scale DCMD apparatus was employed. This apparatus consisted of a horizontal module designed for a flat sheet membrane area of 30.2 cm^2^ and was equipped with two pumps to enable the flow of feed and permeate solutions in counter directions between the membrane module and their respective reservoirs. A schematic of the experimental arrangement is depicted in Figure 2. Fluid circulation was facilitated by a peristaltic pump (Lead Fluid, BT600F, YZ15) with two Easyload II heads (Model 77200-60). Temperature regulation was achieved using both a heating unit (Lauda CS 6-D recirculating bath) and a cooling unit (Lauda-Brinkmann LCK 4929 ECO RE 620 GW Thermostatic Bath and Circulator, Condenser Water), allowing for a wide range of operational temperatures. Continuous monitoring was conducted for the electrical conductivity and temperature of the hot and cold electrolyte solutions and the volume of water transferred to the cold side. The experiments were carried out using a terrace-effect bead SPES/S-MWCNTs bead nanofiber membrane module set up for counter-current flow, with deionized water on the cold side and a saltwater solution (3.5 wt% NaCl) on the hot side. Both pump heads were operated at a consistent flow rate of 150 mL/min, and the temperatures of the fluid streams were maintained at 11 °C for the permeate and 70 °C for the feed, creating a substantial temperature differential of 59 ± 1 °C across the membrane. This differential facilitated the movement of vapor through the hydrophobic membrane, resulting in its condensation on the permeate side. Changes in the weight of the permeate, indicating vapor flux and salt rejection, were measured at 15 min intervals using a precision balance and a conductivity probe, respectively, with conductivity readings taken using a conductivity meter (FiveGo F3, Mettler-Toledo, Columbus, OH, USA).

The permeate flux (L m^−2^ h^−1^) was calculated by the following Equation (1):J = V/A × t (1)
where J is the permeate flux (L m^−2^ h^−1^), V is the volume of permeate (L), A is the effective membrane area (m^2^), and t is the sampling time (h).

The salt rejection (SR) of a membrane can be calculated using the following Equation (2):(2)$$Salt\;Rejection(\%)=(1-\frac{C_{permeate}}{C_{feed}}\;)\times 100\%$$
where C_permeate_ is the concentration of salt in the permeate stream. C_feed_ is the concentration of salt in the feed stream.

This equation calculates the percentage of salt rejected by the membrane, indicating the membrane’s effectiveness in removing salt from the feed solution.

## 3. Results and Discussion

This research encapsulated S-MWCNTs within SPES, leveraging their expansive surface area and superior thermal characteristics. The focus on hydrophobic engineered nanomaterials (ENMs) for MD has intensified recently. Nevertheless, the emergence of ultra-hydrophobic ENMs for potentially unparalleled salt separation is underscored by our findings, which involved the creation of S-MWCNTs that encapsulated hydrophobic SPES@S-MWCNTs, exhibiting elevated adsorption rejection capacities [51,60]. Following the immobilization of nanoparticles, the viscosity of the SP solution exhibited a linear increase from 2410 mPa·s to 2456 mPa·s, while its electrical conductivity rose from 1.6 µS/cm to 1.75 µS/cm.

MWCNTs-0-containing SPES nanofibers are characterized by their smooth surface and cross-section view with and without any beads in the composite nanofibers, as shown in Figure 3. In contrast, for composites with 0.5% and 1% S-MWCNTs, there is a noticeable reduction in bead size compared to S-MWCNT-0, along with an increase in bead quantity. Specifically, the 1% S-MWCNTs variant exhibits a rough surface texture, distinguished by a significant number of beads within its structure. The diameter of these composite nanotubes is approximately 31 ± 1 nm, and an increase in S-MWCNTs content correlates with a decrease in the size of composite nanofibers. SEM imagery detailed in Figure 3 showcases the membrane surfaces, highlighting how S-MWCNTs concentration adjustments can influence bead formation within the membrane framework. Figure 3 reveals a marked morphological distinction in the SPES nanofibers as the S-MWSNTs ratio escalates. Specifically, membranes enriched with higher S-MWCNTs concentrations display a denser bead distribution within the membrane structure, a phenomenon attributed to the increased surface tension due to a higher contact angle [17,61]. Elevating the MWCNTs ratio gradually transitions the fiber morphology from smooth nanofibers to polymer clusters manifesting as beads whose shapes vary with the S-MWCNTs concentration [54,62].

Since the strong hydrophobicity of the membrane surface facing the hot feed solution is essential for the stable operation of a membrane during the MD process, the CA values of hydrophobic layers prepared by electrospinning solutions with different S-MWCNTs weight ratios were measured, and the results are given in Figure 3. The CA value of the hydrophobic layer incorporated with 0.5% S-MWCNTs and 1% S-MWCNTs particles is determined to be 120 ± 1.5° and 145 ± 2°, respectively, which is attributed to the inherent hydrophobicity of silane-modified S-MWCNTs, while the pure SPES membrane CA is 70 ± 1°. The hydrophobicity can be enhanced by the addition of S-MWCNTs and reaches the largest WCA value of 145 ± 2°, in this experiment. However, an increase in the CA value is observed with a further increase in the S-MWCNTs ratio, which is attributed to more protuberances on the bead surface.

Consequently, under identical electrospinning conditions, the diameters of SPES@S-MWCNTs nanofibers are bigger than those of pure SPES nanofibers. This incorporation of S-MWCNTs into SPES membranes is evident from the SEM micrographs. The addition of MWCNTs to the nanofibers may alter their filtration performance. As indicated in Table 1, the diameters of the nanofibers vary with different concentrations of MWCNTs, ranging from 67.5 ± 10.1 to 83.8 ± 12.14 nm. The inclusion of S-MWCNTs influences not only the diameter but also the pore size and porosity of the nanofiber membranes. An increase in nanofiber diameter leads to smaller pore sizes and higher porosity, as detailed in Table 1. Overall, the presence of MWCNTs tends to enhance the diameter, decrease the pore size, and increase the porosity of the nanofiber membranes [63].

Although thinner membranes generally promote greater water flux due to their reduced resistance to flow, the pure SPES membrane, being thinner, exhibited less influence of porosity on flux enhancement. Conversely, thicker SPES@S-MWCNTs nanofiber membranes demonstrated increased flux attributed to significantly higher porosity compared to SPES nanofiber membranes. Nevertheless, despite their capacity for higher flux, thinner membranes with substantial porosity may sacrifice selectivity and structural integrity, factors that are vital depending on the intended application.

### 3.1. Membrane Characterization

Figure 4a,b display the FTIR spectra for both the unaltered SPES and the modified ENMs. Peaks at 1066 and 1181 cm^−1^ correspond to the symmetric stretching vibrations of O=S=O, attributable to the presence of SO_2_ groups within the polymer’s macromolecular structure, as cited in reference [2]. Furthermore, the spectral bands at 1038 cm^−1^ and 1300 cm^−1^ are indicative of the stretching vibrations associated with –SO_3_H groups, confirming SPES’s incorporation in the ENM. Acidic treatment of MWCNTs using H_3_PO_4_ and H_2_SO_4_ leads to the introduction of hydroxyl (–OH) groups on their surface, as evidenced by distinct peaks in the 1724 cm^−1^ range, which are attributed to the C=O stretching vibrations of the COOH groups on the MWCNTs, according to reference [50]. Moreover, the detection of peaks at 821 cm^−1^, 1082 cm^−1^, and 1193 cm^−1^ corresponding to C-F, alongside the peak at 2270 cm^−1^ for N–H stretching and 1654 cm^−1^ for C=O, suggests C-OH stretching vibrations in the diverse chemical environments of silane, as referenced in [64]. Despite these observations, it is clear that sulfonic acid groups, silane, and MWCNTs have been successfully integrated into the polymer matrix.

### 3.2. X-ray Diffraction

The X-ray diffraction (XRD) pattern for MWCNTs is depicted in Figure 4c, revealing the identification of carbon with a characteristic diffraction peak at a 2θ value of 25°, corresponding to the (002) plane, as referenced in [65]. Furthermore, as illustrated in Figure 4c, the observed diffraction peaks at 2θ angles of 33.4°, 47.5°, and 58.4° are assignable to the d(311), d(400), and d(511) diffraction indices, respectively. These peaks substantiate the existence of a face-centered cubic structure within the MWCNTs that are embedded on the SPES surface [65].

### 3.3. Tensile Strength Analysis

The mechanical attributes of engineered nanofiber membrane nanomaterials are essential for their application in membrane distillation research. The mechanical strength of pure SPES and SPES incorporated with S-MWCNTs at concentrations of 0.5% and 1% was assessed using stress–strain curves, as depicted in Figure 4d. Pure SPES nanofiber membranes exhibited notable mechanical strength, with a tensile strength of 5.5 ± 0.9 MPa and an elongation at a break of 69.7%. Adding S-MWCNTs to SPES marginally raised the tensile strength to 5.8 ± 1 MPa while decreasing the elongation to 50.1% for nanofiber membranes containing 1% S-MWCNTs, indicating an interaction between the S-MWCNTs and the macromolecular chains of SPES. Conversely, the 0.5% S-MWCNTs variant demonstrated superior tensile strength and elongation compared to the 1% variant. It is noted that an increase in the concentration of MWCNTs tends to diminish both the tensile strength and elongation of the nanofiber membranes.

### 3.4. Direct Contact Membrane Distillation Performance

Following the adjustment of parameters, the SPES nanofiber membrane, in both its unadulterated and bead-incorporated forms, was integrated into a DCMD module, as depicted in Figure 2. An evaluative study contrasting the efficacy of the DCMD process utilized the SPES nanofiber membrane against its counterpart incorporation S-MWCNT SPES nanofiber membrane. As demonstrated in Figure 5a, each nanofiber membrane showcased significantly high initial flux rates, spanning from 48.4 L m^−2^ h^−1^ for the incorporated SPES to 87.3 L m^−2^ h^−1^ for the 1% S-MWCNTs-enhanced SPES, despite experimental variations. Owing to its augmented porosity, the SPES nanofiber membrane treated with S-MWCNTs manifested a slightly superior flux in comparison to its unmodified counterpart. During a testing period of 5.4 h, a progressive flux reduction was observed in both membranes, a phenomenon potentially attributable to time-dependent pore wetting. It is of particular interest that the flux decrement for the S-MWCNT-enhanced SPES membrane was relatively minimal, with only a 10% decrease, as opposed to a 41% decline observed in the untreated SPES membrane. This performance gap may be ascribed to the diminished porosity and smaller pore dimensions typical of the conventional membrane. Concurrently, the S-MWCNTs-incorporated SPES membrane maintained an extraordinarily high solute rejection rate of 99.9%, in contrast to a gradual reduction to 95.2% for the unaltered PVDF membrane, as shown in Figure 5b. However, as filtration progressed, a continuous decrease in permeate flux was noted, indicative of potential membrane fouling, concentration polarization, and possibly cake layer formation, which collectively hinder molecular transit through the membrane for the SPES nanofiber membrane. Oppositely, the SPES@S-MWCNTs nanofiber membrane showed excellent performance during the experiment. Specifically, membrane fouling seemed to significantly contribute, with accumulating particles or dissolved substances gradually blocking the membrane’s pores or surface area. Accordingly, these findings indicate that the fabricated membrane outperforms the standard SPES nanofiber membrane, attributable to its material composition with a lower surface energy. This enhanced performance is primarily due to the larger pore sizes of the S-MWCNTs treated membranes relative to the standard SPES nanofiber membrane, while its remarkable rejection rate is due to its superhydrophobic properties.

### 3.5. Comparison with Literature

Moreover, it is pertinent and intriguing to juxtapose these results with those documented in the literature concerning comparable membranes with beads as well as with state-of-the-art SPES@S-MWCNTs nanofiber membranes. Notably, the bead nanofiber membrane prepared herein showcases superior performance compared to the limited number of other membranes documented in the literature (Table 2).

## 4. Conclusions

This study on bead-containing hydrophobic SPES@S-MWCNT nanofiber membranes for DCMD concludes that these membranes exhibited exceptional performance characteristics, including a high water flux of 87.3 L m^−2^ h^−1^ and excellent salt rejection of 99.8% for SPES@S-MWCNTs (1%), compared to 48.4 L m^−2^ h^−1^ and 95.2% for pure SPES, underscoring their potential for revolutionizing water purification processes. The integration of bead into the nanofibers significantly enhances the efficiency of distillation operations, making this approach highly advantageous over traditional methods in terms of energy efficiency. Future research should focus on scalability and the longevity of the membrane under varied operational conditions to further validate its commercial viability. The promising results open avenues for the application of these membranes in industrial-scale desalination and wastewater treatment, marking a significant advancement in sustainable water management technologies.

## Figures and Tables

**Figure 1 membranes-14-00120-f001:**
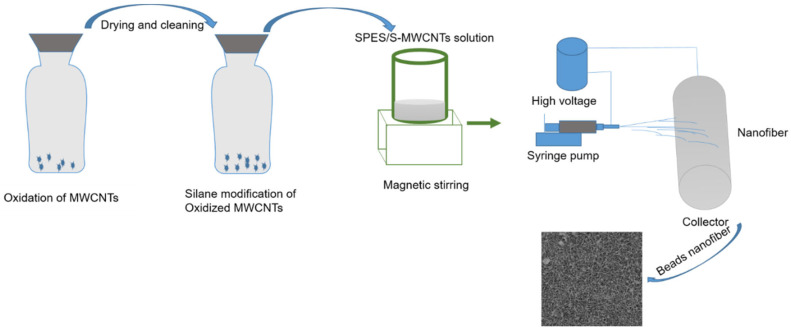
The fabrication process of SPES/MWCNT nanofiber.

**Figure 2 membranes-14-00120-f002:**
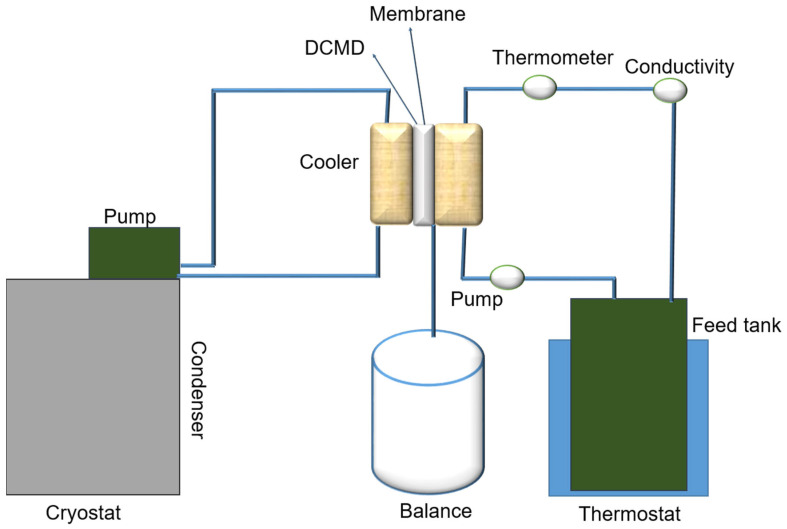
A diagrammatic representation of the DCMD process.

**Figure 3 membranes-14-00120-f003:**
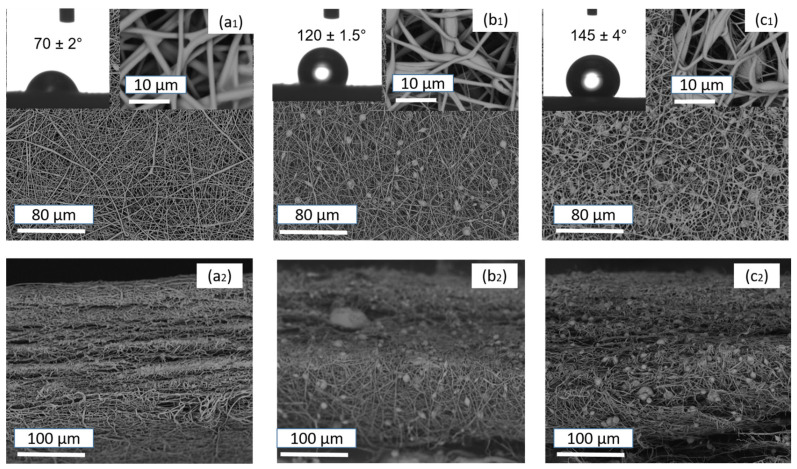
SEM micrographs of top and cross-section view of (**a_1_**,**a_2_**) pristine SPES, (**b_1_**,**b_2_**) SPES@S-MWCNTs (0.5%), and (**c_1_**,**c_2_**) SPES@S-MWCNTs (1%), as well as the water contact angle (WCA) for each membrane.

**Figure 4 membranes-14-00120-f004:**
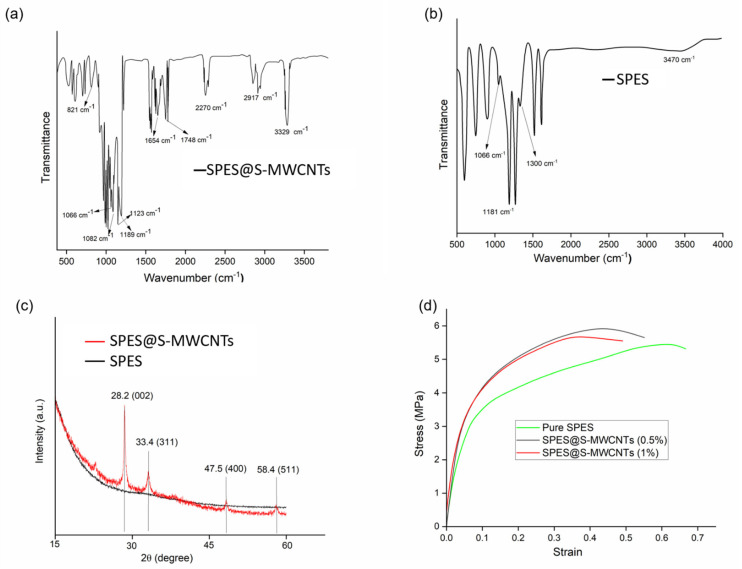
FTIR spectra of SPES and modified SPES@S-MWCNTs (**a**,**b**) and XRD spectra of SPES and modified SPES@S-MWCNTs (**c**). (**d**)Tensile strength for pure SPES and SPES@S-MWCNTs (−0.5 and −1%).

**Figure 5 membranes-14-00120-f005:**
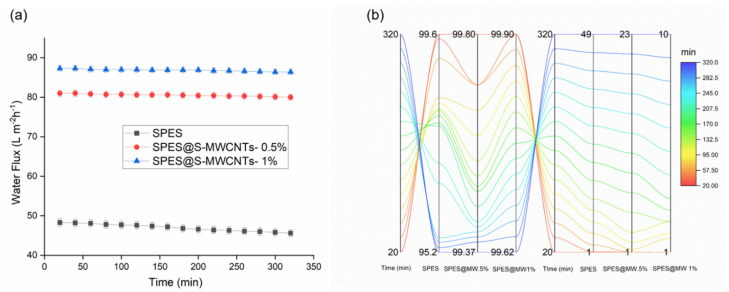
Comparative performances of the SPES and SPES@MWCNTs- 0.5% and SPES@MWCNTs- 1% during 4 h duration DCMD process. (**a**) Average flux; (**b**) salt rejection and conductivity of permeate.

**Table 1 membranes-14-00120-t001:** Physical properties of the SPES and SPES@S-MWCNT solution and bead nanofiber membranes.

Sample	Viscosity (mPa S^−1^)	Electric Conductivity (µS cm^−1^)	Diameter (nm)	Thickness (mm)	Pore Size (µm)	Porosity (%)
SPES	2410	1.6	67.5 ± 10.01	0.4	4.75 ± 0.9	64.4
SPES@S-MWCNTs (0.5%)	2461	1.7	75.8± 11.54	0.45	4.3 ± 0.8	73.9
SPES@S-MWCNTs (1%)	2456	1.75	83.8± 12.14	0.5	3.9 ± 0.7	79.1

**Table 2 membranes-14-00120-t002:** Summary of the DCMD performances of the as-prepared SPES@MWCNTs nanofiber membrane and comparison with pure SPES nanofiber and surfaced modified SPES@MWCNTs nanofiber membranes.

Material	Membrane Preparation Process	CA (°)	∆T (°C)	NaCL Concentration (wt%)	Flux (L m^−2^ h^−1^)	Rejection (%)	Reference
PAN/PS/PMDS	Electrospinning	148.5	40	3.5	27.7	100	[12]
PVDF	CF4 plasma (15 min)/electrospinning	148.5	40	15.354 mg L^−1^	15.3	100	[64]
PVDF-nanofiber	Electrospinning	148	35	100 g L^−1^	10.5	99.99	[17]
CNT/PcH membrane	Electrospinning	158.5	50	70 g L^−1^	29.5	100	[66]
PVDF	Fluorination	155	40	Seawater	19.5	99.7	[67]
PVDF	NIPS	155.3	55	3.5	54.5	99.98	[68]
SPES@MWCNTs	Fluorination/Electrospinning	145	50	3.5	87.3	99.8	This work
Pure SPES	Electrospinning	70	50	3.5	48.4	95.2	This work

## Data Availability

The raw data supporting the conclusions of this article will be made available by the authors on request.

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
