# Peer review of "Bead-Containing Superhydrophobic Nanofiber Membrane for Membrane Distillation"

_membranes, 2024, doi:10.3390/membranes14060120_

Round 1

Reviewer 1 Report

Comments and Suggestions for Authors

This manuscript “Beads containing super hydrophobic nanofiber membrane for 2 membrane distillation” deals with the preparation of membrane for membrane distillation. Although extensive work has been performed, several points must be improved before acceptance of this manuscript.

1)     It is not clear from sections Abstract and Introduction that MWCNTs are added to the membranes as modifiers. This information should be reflected in the abstract, and in the Introduction indicate what other modifiers are used and where else MWCNT is used.

2)     What is "WCA"? There is no decoding in the text of the manuscript.

3)     The novelty of this research is needed in the last paragraph of the introduction.

4)     The last paragraph of the introduction does not list all the methods that have been used to analyze membranes.

5)     The success of the oxidation should be confirmed. It is necessary to compare the characteristics of the original MWCNTs and the oxidized ones.

6)     What amount of MWCNTs was introduced into the membrane matrix and why? This information should be added in the section “2.1. Oxidation process of MWCNTs”

7)     Compliance with the template throughout the manuscript should be checked. For example font and section size for “2.1. Oxidation process of MWCNTs” and “2.2. Electrospun membrane fabrication”

8)     The manuscript uses various abbreviations, such as membrane distillation (MD) and direct contact membrane distillation (DCMD). Should be done uniformly

9)     What is the effective membrane area?

10) How many times has the experiment been conducted to study transport properties? Errors should be given.

11) In Figure 3, the letter a is missing from the signature.

12) Section names should not contain abbreviations.

13) It is necessary to analyze and describe the data in Table 4.

14) The "Conclusions" section should reflect all the results obtained in the study.

Comments on the Quality of English Language

Minor editing of English language required

Author Response

Reviewer 1.

  1. It is not clear from sections Abstract and Introduction that MWCNTs are added to the membranes as modifiers. This information should be reflected in the abstract, and in the Introduction indicate what other modifiers are used and where else MWCNT is used.

Answer:  Thank you for your remarks. We added a clear description of the addition of MWCNTs in the abstract and introduction.

Abstract

This study introduces an innovative approach to enhancing membrane distillation (MD) performance by developing beads containing superhydrophobic sulfonated polyethersulfone (SPES) nanofibers with S-MWCNTs. By leveraging SPES’s inherent hydrophobicity and thermal stability, combined with a nanostructured fibrous configuration, we engineered beads designed to optimize the MD process for water purification applications. Here, oxidized hydrophobic S-MWCNTs were dispersed in a SPES solution at concentrations of 0.5% and 1.0% by weight. These bead membranes are fabricated using a novel electrospinning technique, followed by a post-treatment with the hydrophobic polyfluorinated grafting agent to augment nanofiber membrane surface properties, thereby achieving superhydrophobicity with a water contact angle (WCA) of 145 ± 2 o and a higher surface roughness of 512 nm. The enhanced membrane demonstrated a water flux of 87.3 Lm−2h−1 and achieved nearly 99% salt rejection efficiency at room temperature, using a 3 wt% sodium chloride (NaCl) solution as the feed. The results highlight the potential of superhydrophobic SPES nanofiber beads in revolutionizing MD technology, offering a scalable, efficient, and robust membrane for salt rejection.

Introduction

SPES nanofiber membranes have been extensively researched for heavy metal removal due to their exceptional thermal and mechanical properties as well as their chemical resistance [2]. However, SPES nanofiber membrane may not be ideal for certain filtration experiments due to a few key limitations. The increased hydrophilicity, while beneficial for water permeability, can lead to issues such as excessive swelling, reduced mechanical stability, and enhanced fouling susceptibility, which compromises membrane integrity and efficiency. Therefore, several methods have been employed to improve the nanofiber membrane strength, such as surface modifications, particle incorporation, and graft-copolymerization. On the other hand, Multi-walled carbon nanotubes (MWCNTs) are utilized in membrane filtration due to their exceptional mechanical strength, high chemical stability, and remarkable surface area, which enhance mem-brane durability, efficiency, and selectivity [39, 40]. Their nanoscale structure improves water permeability while effectively blocking larger contaminants. Additionally, the surface of MWCNTs can be chemically modified to target specific pollutants, and their potential antimicrobial properties help reduce biofouling, making them ideal for advanced filtration applications in water treatment, gas separation, and more. The inclusion of MWCNTs in polymeric solutions has been reported to significantly enhance filtration efficiency [41].

2)     What is "WCA"? There is no decoding in the text of the manuscript.

Answer: Thank you for your remarks. We apologize for this mistake. We added the decoding of the WCA.

Water contact angle (WCA)

3)     The novelty of this research is needed in the last paragraph of the introduction.

Answer: Thank you for your remarks.  The novelty of this research has been added in the last paragraph of the introduction.

In the presented study, a novel category of SPES@S-MWCNts beads nanofiber membranes made from SPES polymer, designed for MD was developed using a co-electrospinning apparatus. This apparatus features a needle capable of processing SPES solutions with varying concentrations of S-MWCNTs, enabling the production of SPES beads nanofiber membranes that consist of two distinct types of nanofibers pure SPES membrane and beads formation SPES@S-MWCNts, each exhibiting unique morphological characteristics.

4)     The last paragraph of the introduction does not list all the methods that have been used to analyze membranes.

Answer: Thank you for your remarks. All tested methods have been described in the last paragraph of the introduction.

      In the presented study, a novel category of SPES@S-MWCNTs beads nanofiber membranes made from SPES polymer, designed for MD was developed using a co-electrospinning apparatus. This apparatus features a needle capable of processing SPES solutions with varying concentrations of S-MWCNTs, enabling the production of SPES beads nanofiber membranes that consist of two distinct types of nanofibers (pure SPES membrane and beads formation SPES@S-MWCNts, each exhibiting unique morphological characteristics. Adjusting the ratios of polymer solution solvents and the operational parameters in the fabrication process altered the traditional nanofiber morphology of SPES nanofiber membranes to a beaded structure with micron-sized beads. The density and shape of the beads in the membranes were examined using scanning electron microscopy (SEM), Fourier Transform Infrared Spectroscopy (FTIR), X-ray diffraction (XRD), and tensile strength, alongside evaluations of WCA. The membranes’ efficacy was evaluated in DCMD mode, measuring water flux and salt rejection with their performance benchmarked against another nanofiber membrane.

5)     The success of the oxidation should be confirmed. It is necessary to compare the characteristics of the original MWCNTs and the oxidized ones.

Answer: Thank you for your remarks. The author confirmed that pure MWCNTs were not utilized due to leaching concerns. Consequently, an oxidation process was employed to produce hydroxyl-functionalized MWCNTs (MWCNTs-OH).

6)     What amount of MWCNTs was introduced into the membrane matrix and why? This information should be added in the section “2.1. Oxidation process of MWCNTs”

Answer: Thank you for your remarks. We added this in section 2.2. Electrospun membrane fabrication.

MWCNTs at 0.5% and 1% concentrations were considered optimum for membrane filtration due to their unique properties. At these concentrations, MWCNTs effectively improve the hydrophilicity of membranes, leading to increased water flux and decreased fouling without compromising the structural integrity of the membrane. Higher concentrations might lead to agglomeration and pore clogging, reducing efficiency, while lower concentrations may not provide significant enhancements. Thus, 0.5% and 1% are optimal for balancing performance with cost and material handling.

7)     Compliance with the template throughout the manuscript should be checked. For example font and section size for “2.1. Oxidation process of MWCNTs” and “2.2. Electrospun membrane fabrication”

Answer: Thank you for your comments. In the manuscript, the author has reviewed both sections.

8)     The manuscript uses various abbreviations, such as membrane distillation (MD) and direct contact membrane distillation (DCMD). Should be done uniformly

Answer: Thank you for your remarks. In the manuscript, the author has corrected.

9)     What is the effective membrane area?

Answer: Thank you for your remarks.  The effective membrane area typically refers to the surface area of a membrane that is actively involved in the process of separation, filtration, or other membrane-based processes.

10) How many times has the experiment been conducted to study transport properties? Errors should be given.

Answer: Thank you for your remarks. The author verified that each experiment was conducted five times, and error bars have been included in response to the reviewer's request.

11) In Figure 3, the letter is missing from the signature.

Answer: Thank you for your remarks. In the manuscript, the author has corrected.

12) Section names should not contain abbreviations.

Answer: Thank you for your remarks. In the manuscript, the author has corrected.

13) It is necessary to analyze and describe the data in Table 4.

Answer: Thank you for your remarks. The current manuscript does not contain Table 4.

14) The "Conclusions" section should reflect all the results obtained in the study.

Answer:

      The study on beads containing hydrophobic SPES@S-MWCNTs nanofiber membranes for DCMD concludes that these membranes exhibited exceptional performance characteristics, including high water flux for SPES@S-MWCNTs (1%) was 87.3 L m-2h-1, and excellent salt rejection 99.8 %, compared to pure SPES was 48.4 L m-2h-1  and salt rejection 95.2 % which underscore their potential for revolutionizing water purification processes.

Reviewer 2 Report

Comments and Suggestions for Authors

This manuscript describes the fabrication and use of a hydrophobic composite membrane for use in a direct-contact membrane distillation system. This study proposes a scalable method of electrospinning sulfonated polyethersulfone (SPES) with multiwalled carbon nanotubes. These membranes exhibit good performance characteristics, including high water flux and high salt rejection capabilities. This research has the potential to greatly benefit researchers and membrane manufacturers in this field trying to produce cost effective membranes for commercial MD to be feasible. However, there are many missing information in this manuscript. It is recommended to publish upon major revisions.

Detailed comments:

1.     Line 137. I would suggest authors include more membrane characterization since the highlight of this paper is the scalable method of electrospinning SPES with MWCNT and using them in MD. It is not sufficient to base feasibility on just the flux and rejection alone. There should be tests on mechanical strength via tensile test, pore size distribution, SEM image of the membrane’s cross-section, etc.

2.     Line 262. The authors mentioned that the performance gap between the conventional SPES membrane and the SPES with added MWCNT may be ascribed to the diminished porosity and smaller pore dimensions typical of the conventional membrane. However, there was no data of the porosity and pore size of the membranes fabricated in this manuscript. Furthermore, there are many other factors contributing to the flux in a typical DCMD process such as membrane thickness, thermal conductivity of the membrane, etc, but they were not discussed in this manuscript. This lowers the confidence of readers on the authors claim on the improved performance based on hydrophobicity. Firstly, I strongly suggest the authors to include thickness and pore size distribution measurements. Secondly, SEM images of the cross-section should be included to show the membrane morphology along the thickness of the membrane. Thirdly, mechanical strength of the membrane should be tested using a tensile tester to give insights on its feasibility for long term usage compared to other commercially available membrane. Furthermore, I suggest that if inlets and outlets temperatures of the DCMD experiments are available they should be briefly discussed to allow readers to know how the thermal conductivity of the membrane are changing due to addition of MWCNT with SPES. Lastly, I suggest that the authors look at basic MD flux prediction equations and use the experimental data they have to predict the flux and check if the predicted % increase matches the experimental % increase.

3.     Line 282. The list used for comparison did not include many key information such as thickness, membrane pore size, feed and distillate temperatures, are they all operated using DCMD configurations, etc. I do understand that a fair comparison might be difficult across the different research, but such information that affects flux should be present for readers to make sense of the conclusions the authors are proposing. The authors should also include that the DCMD set up has the highest heat loss across the membrane and that at similar operating conditions and membrane should produce lesser flux compared to other configurations.

Comments on the Quality of English Language

The quality of English language is acceptable and readers can understand the conclusion the authors are trying to put across.

Author Response

Reviewer 2

  1. Line 137. I would suggest authors include more membrane characterization since the highlight of this paper is the scalable method of electrospinning SPES with MWCNT and using them in MD. It is not sufficient to base feasibility on just the flux and rejection alone. There should be tests on mechanical strength via tensile test, pore size distribution, SEM image of the membrane’s cross-section, etc.

Answer:

Following the immobilization of nanoparticles, the viscosity of the SP solution exhibited a linear increase from 2410 mPa·s to 2456 mPa·s, while its electrical conductivity rose from 1.6 µS/cm to 1.75 µS/cm. Consequently, under identical electrospinning conditions, the diameters of SPES@S-MWCNTs nanofibers are bigger than those of pure SPES nanofibers. This incorporation of S-MWCNTs into SPES membranes is evident from the SEM micrographs. The addition of MWCNTs to the nanofibers may alter their filtration performance. As indicated in Table 1, the diameters of the nanofibers vary with different concentrations of MWCNTs, ranging from 67.5 ± 10.1 to 83.8 ± 12.14 nm. The inclusion of S-MWCNTs influences not only the diameter but also the pore size and porosity of the nanofiber membranes. An increase in nanofiber diameter leads to smaller pore sizes and higher porosity, as detailed in Table 1. Overall, the presence of MWCNTs tends to enhance the diameter, smaller pore size, and higher porosity of the nanofiber membranes.

Table 1. Physical properties of the SPES and SPES@S-MWCNTs solution and nanofiber membranes

Sample

Viscosity (mPa S-1)

electric conductivity (µS cm-1)

Diameter (nm)

Pore size

(µm)

Porosity

(%)

SPES

2410

1.6

67.5 ± 10.01

4.75 ± 0.9

64.4

SPES@S-MWCNTs (0.5%)

2461

1.7

75.8± 11.54

4.3 ± 0.8

73.9

SPES@S-MWCNTs (1%)

2456

1.75

83.8± 12.14

3.9 ± 0.7

79.1

Tensile strength analysis

The mechanical attributes of engineered nanofiber membranes are essential for their application in membrane distillation research. The mechanical strength of pure SPES and SPES incorporated with S-MWCNTs at concentrations of 0.5% and 1% was assessed using stress-strain curves depicted in Figure 4d. Pure SPES nanofiber membranes exhibited notable mechanical strength, with a tensile strength of 5.5 ± 0.9 MPa and an elongation at a break of 69.7%. Adding S-MWCNTs to SPES marginally raised the tensile strength to 5.8 ± 1 MPa while decreasing the elongation to 50.1% for nanofiber membranes containing 1% S-MWCNTs, indicating an interaction between the S-MWCNTs and the macromolecular chains of SPES. Conversely, the 0.5% S-MWCNTs variant demonstrated superior tensile strength and elongation compared to the 1% variant. It is noted that an increase in the concentration of MWCNTs tends to diminish both the tensile strength and elongation of the nanofiber membranes.

  1. Line 262. The authors mentioned that the performance gap between the conventional SPES membrane and the SPES with added MWCNT may be ascribed to the diminished porosity and smaller pore dimensions typical of the conventional membrane. However, there was no data of the porosity and pore size of the membranes fabricated in this manuscript. Furthermore, there are many other factors contributing to the flux in a typical DCMD process such as membrane thickness, thermal conductivity of the membrane, etc, but they were not discussed in this manuscript. This lowers the confidence of readers on the authors claim on the improved performance based on hydrophobicity. Firstly, I strongly suggest the authors to include thickness and pore size distribution measurements. Secondly, SEM images of the cross-section should be included to show the membrane morphology along the thickness of the membrane. Thirdly, mechanical strength of the membrane should be tested using a tensile tester to give insights on its feasibility for long term usage compared to other commercially available membrane. Furthermore, I suggest that if inlets and outlets temperatures of the DCMD experiments are available they should be briefly discussed to allow readers to know how the thermal conductivity of the membrane are changing due to addition of MWCNT with SPES. Lastly, I suggest that the authors look at basic MD flux prediction equations and use the experimental data they have to predict the flux and check if the predicted % increase matches the experimental % increase.

Answer: 

Thanks for your comments. The author confirmed that although thinner membranes generally promote greater water flux due to their reduced resistance to flow, the purely SPES membrane, being thinner, exhibited less influence of porosity on flux enhancement. Conversely, thicker SPES@S-MWCNTs nanofiber membranes demonstrated increased flux attributed to significantly higher porosity compared to SPES nanofiber membranes. Nevertheless, despite their capacity for higher flux, thinner membranes with substantial porosity may sacrifice selectivity and structural integrity, factors that are vital depending on the intended application.

  1. Line 282. The list used for comparison did not include many key information such as thickness, membrane pore size, feed and distillate temperatures, are they all operated using DCMD configurations, etc. I do understand that a fair comparison might be difficult across the different research, but such information that affects flux should be present for readers to make sense of the conclusions the authors are proposing. The authors should also include that the DCMD set up has the highest heat loss across the membrane and that at similar operating conditions and membrane should produce lesser flux compared to other configurations.

Answer: Thanks for your comments. The author confirmed that all information has been added to the manuscript as per your comments.

Round 2

Reviewer 1 Report

Comments and Suggestions for Authors

The article can be accepted in present form

Author Response

Thank you

Reviewer 2 Report

Comments and Suggestions for Authors

This manuscript describes the fabrication and use of a hydrophobic composite membrane for use in a direct-contact membrane distillation system. This study proposes a scalable method of electrospinning sulfonated polyethersulfone (SPES) with multiwalled carbon nanotubes. These membranes exhibit good performance characteristics, including high water flux and high salt rejection capabilities. This research has the potential to greatly benefit researchers and membrane manufacturers in this field trying to produce cost effective membranes for commercial MD to be feasible.

The authors addressed all the comments brought up the first round of review. The revision is satisfactory. However, there are some questions that might have been misunderstood the first time round. It is recommended to publish upon minor revisions.

Detailed comments:

1.    Table 1 Line 277. The authors gave the diameter of the fibres instead of the total thickness of the membrane. Please include the thickness of the membranes for completeness.

2.    Table 2 Line 367. The authors did not include the configuration that the references cited have used (i.e. DCMD, VMD, etc.) and what distillate and feed temperatures they are operating at? These are important parameters that affect flux and they should be included.

Author Response

Response to comments: Reviewer 2:

  1. Table 1 Line 277. The authors gave the diameter of the fibres instead of the total thickness of the membrane. Please include the thickness of the membranes for completeness.

 Answer: Thank you for your remarks. We apologize for this mistake. We added the thickness in the manuscript

    Table 1. Physical properties of the SPES and SPES@S-MWCNTs solution and beads nanofiber membranes

Sample

Viscosity (mPa S-1)

electric conductivity (µS cm-1)

Diameter (nm)

Thickness

(mm)

Pore size

(µm)

Porosity

(%)

SPES

2410

1.6

67.5 ± 10.01

0.4

4.75 ± 0.9

64.4

SPES@S-MWCNTs (0.5%)

2461

1.7

75.8± 11.54

0.45

4.3 ± 0.8

73.9

SPES@S-MWCNTs (1%)

2456

1.75

83.8± 12.14

0.5

3.9 ± 0.7

79.1

  1. Table 2 Line 367. The authors did not include the configuration that the references cited have used (i.e. DCMD, VMD, etc.) and what distillate and feed temperatures they are operating at? These are important parameters that affect flux and they should be included.

Answer: Thank you for your remarks. We apologize for this mistake. We added the feed and permeate temperature differences in the table of main manuscript.

Table 2. Summary of the DCMD performances of the as-prepared SPES@MWCNTs nanofiber membrane and comparison with pure SPES nanofiber and surfaced modified SPES@MWCNTs nanofiber membranes.

Material

Membrane preparation process

CA

(°)

∆T

      (á´¼C)

NaCL concentration (wt%)

Flux (L m-2h-1)

Rejection (%)

Reference

PAN/PS/PMDS

Electrospinning

148.5

40

3.5

27.7

100

[38]

PVDF

CF4 plasma (15 min)/ electrospinning

148.5

40

15.354 mg L-1

15.3

100

[66]

PVDF-nanofiber

Electrospinning

148

35

100 g L-1

10.5

99.99

[37]

CNT/PcH membrane

Electrospinning

158.5

50

70  g L-1

29.5

100

[68]

PVDF

Fluorination

155

40

Seawater

19.5

99.7

[69]

PVDF

NIPS

155.3

55

3.5

54.5

99.98

[70]

SPES@MWCNTs

Fluorination/ Electrospinning

145

50

3.5

87.3

99.8

This work

Pure SPES

Electrospinning

70

50

3.5

48.4

95.2

This work